# DP-RAE: A Dual-Phase Merging Reversible Adversarial Example for Image Privacy Protection

## ABSTRACT

In digital security, Reversible Adversarial Examples (RAE) blend adversarial attacks with Reversible Data Hiding (RDH) within images to thwart unauthorized access. Traditional RAE methods, however, compromise attack efficiency for the sake of perturbation concealment, diminishing the protective capacity of valuable perturbations and limiting applications to white-box scenarios. This paper proposes a novel Dual-Phase merging Reversible Adversarial Example (DP-RAE) generation framework, combining a heuristic black-box attack and RDH with Grayscale Invariance (RDH-GI) technology. This dual strategy not only evaluates and harnesses the adversarial potential of past perturbations more effectively but also guarantees flawless embedding of perturbation information and complete recovery of the original image. Experimental validation reveals our method's superiority, secured an impressive 96.9% success rate and 100% recovery rate in compromising black-box models. In particular, it achieved a 90% misdirection rate against commercial models under a constrained number of queries. This marks the first successful attempt at targeted black-box reversible adversarial attacks for commercial recognition models. This achievement highlights our framework's capability to enhance security measures without sacrificing attack performance. Moreover, our attack framework is flexible, allowing the interchangeable use of different attack and RDH modules to meet advanced technological requirements.

## CCS CONCEPTS

• **Security and privacy** → **Privacy protections**; *Systems security*; • **Computing methodologies** → **Computer vision tasks**; **Computer vision tasks**.

## KEYWORDS

Adversarial attack, Privacy protection, Black-box attack

## 1 INTRODUCTION

Deep neural networks (DNNs) have taken several domains [10, 16, 18, 21, 34], by storm due to their unique expressive capabilities and superior performance. However, with the rapid and unregulated expansion of DNNs, human concerns regarding privacy and security continuously intensify [2, 4, 28, 30, 41]. Numerous malicious commercial entities illicitly harvest user privacy to achieve their

Permission to make digital or hard copies of all or part of this work for personal or classroom use is granted without fee provided that copies are not made or distributed for profit or commercial advantage and that copies bear this notice and the full citation on the first page. Copyrights for components of this work owned by others than the author(s) must be honored. Abstracting with credit is permitted. To copy otherwise, or republish, to post on servers or to redistribute to lists, requires prior specific permission and/or a fee. Request permissions from permissions@acm.org.
*ACMMM, 28 October – 1 November, 2024, Melbourne, AU*
© 2024 Copyright held by the owner/author(s). Publication rights licensed to ACM.
ACM ISBN 978-1-4503-XXXX-X/18/06
https://doi.org/XXXXXXX.XXXXXXX

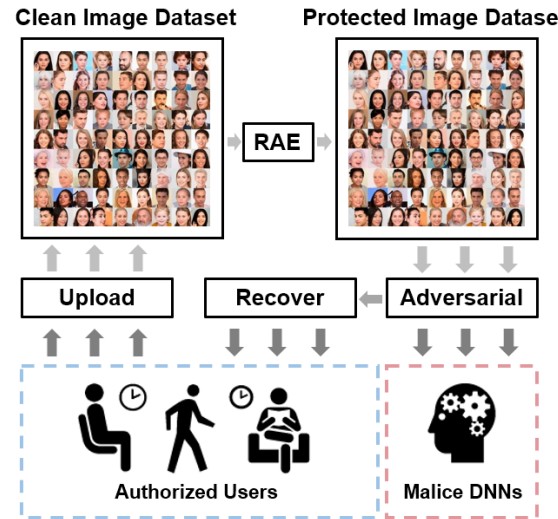

**Figure 1: RAEs thwart malicious DNNs from stealing privacy data and recover image quality via RDH, ensuring authorized users' normal access.**

profit-driven goals. For example, personal data belonging to millions of Facebook users leakage and was analysed by Cambridge Analytica for political advertising [3].

Recent research has exposed DNN vulnerabilities, proposing algorithms to deceive these models with minimal image modifications, thereby inducing incorrect classifications [35, 37, 44, 45, 48]. This paper leverages such adversarial characteristics to develop a novel privacy protection mechanism within social networks. While traditional adversarial attacks can effectively mislead DNNs [7–9, 25, 39], preventing targeted information retrieval and key feature extraction based on DNN models, introducing perturbations to protected images compromises their quality. Hence, there is a pressing need for a privacy-preserving mechanism that upholds both the visual integrity of images and their ability to counteract diverse DNN analyses. Our in-depth research leads to developing Reversible Adversarial Examples (RAE) [24, 43, 46], showcasing their significant capability in confusing DNN classification and analysis while allowing images to be reverted back to their original form. Illustrated in Figure 1, RAEs not only deceive DNNs but also provide a method to revert images to their normal state, introducing an innovative method for protecting image privacy.

In the emerging area of Reversible Adversarial Examples (RAE), existing research is still developing. Xiong *et al.* [43] was the first to apply Reversible Data Hiding (RDH) techniques in black-box attacks by embedding compressed data of perturbations into the images. This approach allows the original images to be recovered. Besides, Their method, by integrating ensemble model techniques

[6], demonstrates the potential of adversarial examples to be applied across different models due to their transferability, yet it falls short in accurately attacking models that have not been previously encountered. On a different note, Zhang *et al.* [46] explored an alternative approach to reverse adversarial examples. They introduced an RGAN, consisting of an attack encoder network and a recovery decoder network, aimed at efficiently producing adversarial examples and then reversing them. While this approach is effective in recovering the original state of images, it underperforms against black-box models.

Given the specialized use cases of RAE, most current RAE research relies on the transferability of white-box attacks for implementing black-box scenarios. Yet, the success of this transferability depends greatly on the strength of the perturbations. The RDH methods used in RAE have strict limits on the perturbation size, creating a design conflict that substantially reduces the effectiveness of RAE against unknown black-box models.

To overcome the inherent challenges for existing RAE methodologies, specifically the tension between executing effective adversarial attacks and adhering to the stringent perturbation constraints of RDH technologies, we propose a Dual-Phase merging Reversible Adversarial Example (DP-RAE) generation framework. This framework adeptly marries the transferability strengths of both white-box and black-box attacks through a bifurcated attack process, significantly improving cross-model applicability. Besides, to address the perturbation storage issues posed by RDH, we propose two novel perturbation optimization techniques: Gradient Quantized Binary Encoding (GQBE) for white-box scenarios and Threshold-Informed Superpixel Attack (TISA) for black-box contexts. The GQBE approach capitalizes on integrated gradient information from models to classify gradient magnitudes into discrete perturbation levels during iterative updates. This stratification reduces the complexity of perturbation data, facilitating its conversion into binary streams for seamless RDH integration. Conversely, if GQBE fails to achieve transferable attacks on uncharted black-box models, TISA is employed to bolster transferability. TISA segments images into superpixel blocks, selecting them randomly to apply uniform perturbations. The impact on model confidence is assessed following each perturbation; a rise in confidence prompts a reversal in the perturbation direction of the targeted superpixel block. Following every set of $m$ perturbations, an analysis of historical alterations identifies three points below the perturbation cap for additional adjustment. The culmination of this meticulously designed process allows for the final perturbations to be stored as binary streams via RDH, thereby ensuring their effective preservation within the RDH framework without compromising image integrity.

In summary, the main contributions of this paper are as follows:

• We propose a novel dual-phase merging framework for RAE generation, significantly enhancing the transferability of adversarial examples to both unknown and commercial black-box models. To the best of our knowledge, our approach is the first successful application of RAE attacks on commercial black-box models.

• We propose the GQBE and TISA as innovative solutions for reversible perturbations with RDH technologies. These methodologies address the critical challenge of maintaining the integrity of adversarial examples while ensuring their reversibility.

• Experimental results affirm the superiority of our attack framework, achieving a 96.9% Attack Success Rate (ASR) for reversible adversarial examples on specific models, with a 100% restoration rate for the recovered images. These outcomes validate the practical feasibility of our approach.

## 2 BACKGROUND

**Adversarial Attacks** are broadly classified into white-box and black-box strategies. White-box attacks entail complete access to the target models' architecture and parameters. Goodfellow *et al.* [11] introduced the Fast Gradient Sign Method (FGSM) in 2014, leveraging single-step gradient updates for generating perturbations:

$$\eta = \epsilon \, \mathrm{sign}\left(\nabla_{\boldsymbol{x}} J(\boldsymbol{\theta}, \boldsymbol{x}, y)\right). \tag{1}$$

Expanding on FGSM, Kurakin *et al.* [20] developed the Basic Iterative Method (BIM) in 2016, enhancing the granularity and attack success rate through iterative small-step perturbations. Dong *et al.* [6] subsequently introduced the Momentum Iterative Fast Gradient Sign Method (MI-FGSM) in 2018, which integrates a momentum component to ensure the attack progresses in a steady direction, thereby improving the precision and reliability of adversarial attack. In contrast, black-box attacks lack direct model knowledge, instead leveraging output observations. The Query-Efficient Boundary-based blackbox Attack (QEBA) [22] estimates decision boundaries via gradient direction, while the Simple Black-box Attack (SimBA) [12] alters inputs based on output changes. Surfree [27] generates perturbations by exploiting classifier decision boundary geometry, showcasing the advancement in adversarial techniques and the nuanced understanding of model vulnerabilities.

**Reversible Data Hiding (RDH)** is a typical technique to extract embedded hidden data from labeled camouflage images. Tian *et al.* [38] pioneered the RDH technique via difference expansion, embedding secret data by enlarging the differences between adjacent pixels. Subsequent research has explored utilizing the histogram properties of images for data hiding [23, 47]. However, conventional RDH methods often induce distortions in the grayscale versions of images, which is important in feature analyses of images. Therefore, our paper adopted the RDH with Grayscale Invariance (RDH-GI) proposed by Hou *et al.* [14], which uses the R and B channels of the color image to embed information and ensure grayscale invariance by adjusting the pixel value of the G channel.

## 3 PROPOSED METHOD

### 3.1 Overview

In this section, we introduce the Dual-Phase Merging Reversible Adversarial Example (DP-RAE) framework, which leverages dual-phase crossover techniques for targeted attacks on black-box models. As shown in Figure 2, the DP-RAE consists of three main components: a white-box attack using Gradient Quantized Binary Encoding (GQBE), a black-box attack via Threshold-Informed Superpixel Attack (TISA), and Reversible Perturbation Embedding and Recovery for maintaining the attack's integrity and reversibility. Initially, GQBE preprocesses images for robust adversarial examples, reducing black-box attack query costs. TISA then intensifies the attack, producing the DP-AE and perturbation matrix. GQBE and TISA

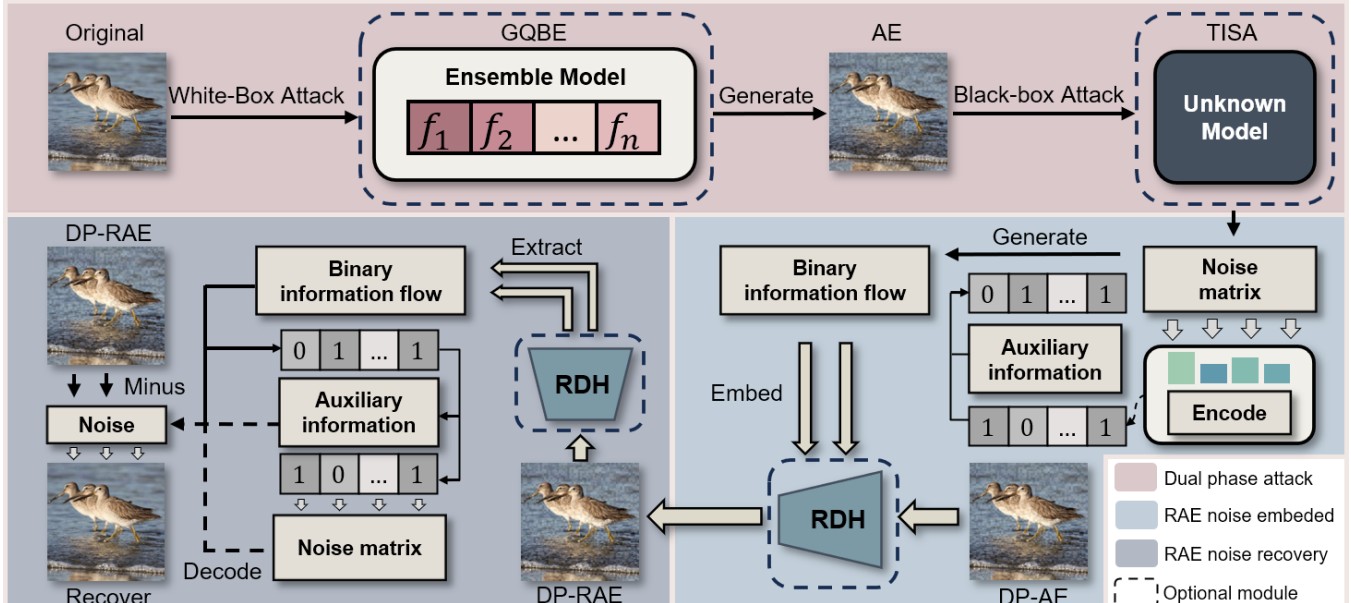

**Figure 2: An overview of our proposed framework.**

can adjust to diverse attack strategies independently. Finally, embedding this matrix and additional data into DP-AE via RDH yields DP-RAE. For restoration, the RDH extracts the hidden information from the DP-RAE, restores the perturbation and recovers the original image.

## 3.2 Gradient Quantized Binary Encoding

Before delving into GQBE, it's essential to address RDH's limitations on information length, making perturbation data compression vital. Our study introduces GQBE, a white-box attack leveraging super-pixels for efficient perturbation compression. This method reduces storage needs while maintaining adversarial effectiveness through gradient smoothing over super-pixels. Consequently, the perturbations retain their capacity to effectively challenge models despite the reduced data footprint.

We denote $x$ as the clean image with the dimensions $C \times H \times W$, $y^{true}$ as the true label, and y as the model $f$ output result: $y = \mathbb{F}_\theta(x)$. Super-pixel size as $h \times w$, $\eta$ represents the perturbation generated by GQBE. The adversarial examples $x'_{adv}$ can be expressed as:

$$x'_{adv} = max(0, min(x + \mathcal{T}(\eta), 1)), \quad (2)$$

where $\mathcal{T}$ is a function designed for dimension expansion and padding. Due to the intervention of super-pixels, $\eta$ essentially acts as a compressed two-dimensional matrix perturbation with dimensions $(\lfloor \mathcal{H}/h \rfloor, \lfloor \mathcal{W}/w \rfloor)$. More specifically, $\eta_{ij}$ represents the perturbation of the super-pixel at position $(i, j)$, and the $\mathcal{T}$ function expands this perturbation to cover the area $(c, i : i + h, j : j + w)$, where $0 \leq i \leq \lfloor \mathcal{H}/h \rfloor, 0 \leq j \leq \lfloor \mathcal{W}/w \rfloor$. We use $\epsilon$ as the unit of perturbation, employing a three-bit code to denote the magnitude of perturbation at $\eta_{ij}$, which signifies the count of unit perturbations.

Then, we dissect how to craft perturbations based on the deviation of the gradients. Initially, the purpose of GQBE is to mislead

classifier $\mathbb{F}_\theta$ through perturbations:

$$\mathbb{F}_\theta(x'_{adv}) = y \neq y^{true}, \ s.t. \|\mathcal{T}(\eta)\|_\infty \leq \epsilon \cdot \mathbb{M} \quad (3)$$

where $\mathbb{M}$ represents the maximum multiplicative factor stored in three bits. To generate $x'_{adv}$, we compute pixel-wise gradients from the loss function, adding perturbations to increase the loss in non-targeted attacks. Given the variability of gradient values across different positions, applying uniform perturbations would lead to varied impacts on the loss function. Recognizing this, we prioritize larger perturbations at points with a more significant influence on the loss function, as these regions are more sensitive to changes in input that substantially affect the final classification decision, potentially making the attack more effective. Consequently, by smoothing the gradients for each super-pixel:

$$\nabla_\eta \mathcal{J}(x, y^{true})_{ij} = \frac{\sum_0^c \sum_0^h \sum_0^w \nabla_x J(x, y^{true})_{ij}}{c \times h \times w}, \quad (4)$$

and obtain the the absolute value matrix $\mathcal{A}$ through $\nabla_\eta \mathcal{J}(x, y^{true})$:

$$\mathcal{A} = \left| \nabla_\eta \mathcal{J}(x, y^{true}) \right|, \quad (5)$$

finally, compute the gradient contribution score $\mathcal{E}$:

$$\mathcal{E}_{ij} = \frac{exp(\mathcal{A}_{ij})}{\sum_{p=0}^{i \times j} exp(\mathcal{A}_p)}, \ s.t. i \in [0, \lfloor \mathcal{H}/h \rfloor], j \in [0, \lfloor \mathcal{W}/w \rfloor] \quad (6)$$

where $\mathcal{E}$ denotes the impact of perturbations at different positions on the loss function, termed as the contribution score of gradients to the deviation in the loss function. Given the varied contributions, the generated perturbation values are quantized into multiple levels. The algorithmic procedure is outlined as Algorithm 1.

It is noteworthy that the GQBE framework exhibits compatibility with a broad spectrum of gradient-based methods, serving as its foundational mechanism. Such versatility enables the dynamic refinement of our attack's potency, ensuring its alignment with the

evolving landscape of adversarial attack methodologies. This paper illustrates the application of the GQBE framework by deploying the ensemble attack strategy within MI-FGSM [6], a paradigmatic example that elucidates our attack process in detail. For ensemble attacks, we fuse the logits of K different models:

$$l(x) = \sum_{k=1}^{K} w_k l_k(x), \tag{7}$$

where $l_k(x)$ represents the logits of the k-th model and $w_k$ as it weight. To decrease the confidence $p(x)$, the loss function $\mathcal{J}$ is defined as:

$$\mathcal{J}(x) = -1_{y^{true}} \cdot log(\frac{exp(l(x,t))}{\sum_{c=1}^{C} exp(l(x,c))}), \tag{8}$$

where $1_{y^{true}}$ is the one-hot encoding of $y^{true}$, $C$ represents the total number of classes and $t$ is the correct category of the image. The detailed procedure of the GQBE attack is methodically delineated in Algorithm 2, providing a comprehensive step-by-step guide to implementing this novel attack strategy.

---

**Algorithm 1:** Gradient Contribution Score

**Input:** Gradient of the super-pixels $\nabla_\eta \mathcal{J}(x, y^{true})$
**Input:** Percentage $\mathcal{PCT}$; unit perturbation $\epsilon$
**Output:** Perturbation $\xi$
Initialize the absolute value matrix $\mathcal{A}$, the sign function $\mathcal{S}$ and the contribution score matrix $\mathcal{E}$;
$\mathcal{A} = \left| \nabla_\eta \mathcal{J}(x, y^{true}) \right|$;
$\mathcal{S} = sign\left( \nabla_\eta \mathcal{J}(x, y^{true}) \right)$;
Compute the contribution score matrix $\mathcal{E}$;
**for** $0 \le i \le \lfloor \mathcal{H}/h \rfloor$ **do**
  **for** $0 \le j \le \lfloor \mathcal{W}/w \rfloor$ **do**
    $\mathcal{E}_{ij} = e^{\mathcal{A}_{ij}}/\sum_{p=0}^{i \times j} e^{\mathcal{A}_p}$;
  **end**
**end**
Obtain the coordinates of the top $\mathcal{PCT}$ values in the $\mathcal{E}$;
Set the values at these positions in $\mathcal{A}$ to 2, and the rest to 1;
$\xi = \epsilon(\mathcal{A} \odot \mathcal{S})$;
return $\xi$

---

## 3.3 Threshold-Informed Superpixel Attack

After integrating preprocessing techniques derived from white-box attacks, the DP-RAE execute a black-box attack on an unexposed model. We utilize the perturbation $\eta$ generated by the Algorithm 2 and clean image $x$ as the input. A direction $q$ is randomly selected by choosing $\eta_{ij}$. Adding a perturbation in the $q$ direction will change the confidence $p$ in the model. If the direction $q$ fails to decrease the $p(y^{true}|x + \mathcal{T}(\eta + q \cdot \epsilon))$, the direction of $q$ will be reversed. For each perturbed point, we document its contribution to the reduction in confidence. Following a predefined number $m$ of perturbation iterations, we identify and select the three points that exhibit the most substantial decrease in model confidence from these documented instances. As the super-pixel blocks of perturbations may have already reached their maximum threshold, augmenting these pairs of perturbations may not directly affect the confidence. Therefore, we excluded these points and only augmented the perturbations for the points that could be added.

---

**Algorithm 2:** Gradient Quantized Binary Encoding

**Input:** The logits of $\mathcal{K}$ classifiers $l_1, l_2, ..., l_k$; ensemble weights $w_1, w_2, ..., w_k$; clean image x; and the true label $y^{true}$
**Input:** Percentage $\mathcal{PCT}$; unit perturbation $\epsilon$; iteration $\mathcal{I}$; and the maximum multiplicative factor $\mathbb{M}$
**Output:** Adversarial examples $x'_{adv}$; Perturbation $\eta$
Initialize $g_0 = 0$, $x'^0_{adv} = x$, $\eta_0$ = zero matrix;
**for** $i = 0$ $to$ $\mathcal{I} - 1$ **do**
  Input $x'^i_{adv}$ and output $l_k(x'^i_{adv})$ for $k = 1, 2, ..., \mathcal{K}$;
  Fuse the logits as $l(x'^i_{adv}) = \sum_{k=1}^{K} w_k l_k(x'^i_{adv})$;
  Get softmax cross entropy loss $\mathcal{J}(x'^i_{adv}, y^{true})$ based on $l(x'^i_{adv})$ and Eq. (8);      ▷ Apply MI
  Smooth the gradient of super-pixels based on Eq. (4) to obtain the gradient: $\nabla_{\eta_i} \mathcal{J}((x'^i_{adv}, y^{true})$;
  Input $\nabla_{\eta_i} \mathcal{J}((x'^i_{adv}, y^{true}), \mathcal{PCT}, \epsilon$ in Algorithm 1 and obtain the output $\xi$;
  Clip $\eta_{i+1}$ to ensure $\|\mathcal{T}(\eta_{i+1})\|_\infty \le \epsilon \cdot \mathbb{M}$;
  $x'^{i+1}_{adv} = max(0, min(x + \mathcal{T}(\eta_{i+1}), 1))$;
**end**
return $x'_{adv}$ and $\eta$

---

## 3.4 Embed And Recover

DP-RAE can utilize RDH-GI technology to embed additional data intricately into adversarial images. Upon generating the adversarial examples with the dual-phase attack mechanism, DP-AE, the perturbation matrix is encoded into a binary information stream. This stream is then meticulously embedded into the adversarial image alongside pertinent auxiliary data, ensuring the integrity of the embedded information while maintaining the adversarial nature of the image. When the recovery of the original image becomes necessary, the hidden information within DP-RAE is extracted using the RDH-GI technology. This enables accurate reconstruction of the perturbation matrix and lossless recovery of the pristine image upon its removal. Through this innovative approach, DP-RAE not only maintains the efficacy of adversarial attacks but also ensures the reversibility of the process, allowing for the seamless restoration of the unmodified original image.

## 4 EXPERIMENTS

### 4.1 Experiment setup

We use ILSVRC2012 [31] as the dataset for our experiments, which is widely used in deep learning and is highly representative and influential. The dataset covers 1000 different categories, and each image is correctly labeled. In terms of model selection, we chose several models to test the migration performance of adversarial attack samples across different models, including Resnet34 (RN-34) [13], Resnet50 (RN-50) [13], Resnet152 (RN-152) [13], DenseNet-121 (DN-121) [17], MobileNet-v2 (Mob-v2) [32], MobileNet-v3 (Mob-v3) [15], VGG-16 [33], VGG-19 [33], AlexNet [19], Inception-v3 (Inc-v3) [36]. To ensure the rigor and fairness of the experiment, we

randomly selected 1000 images from ILSVRC2012, each of which can be reliably classified by the above models.

The parameter settings for DP-RAE are as follows: the super-pixel size is set to 4, the unit perturbation $\epsilon$ is set to 4/255, the number of iterations for GQBE is set to 10, and the $\mathcal{PCT}$ is set to 50%. As for TISA, the attack is set to 3000 iterations, with an additional adjustment added every ten perturbations. All experiments were performed on the NVIDIA A40 GPU.

To rigorously assess the efficacy of DP-RAE, we focus on three key dimensions: attack ability, restoration quality, and visual integrity. We evaluate DP-AE and DP-RAE using established benchmarks for image quality assessment, specifically Peak Signal-to-Noise Ratio (PSNR) and Structural Similarity Index (SSIM) [40]. In addition, we evaluate the model's susceptibility to adversarial attacks using the Attack Success Rate (ASR), which measures the probability of adversarial examples causing misclassifications.

## 4.2 Attack ability

To demonstrate the superior performance of DP-RAE, we compared it to a comprehensive set of adversarial attack methods. These include classical white-box attacks such as the FGSM [11], BIM [20], and Projected Gradient Descent (PGD) [26], alongside advanced black-box techniques like the SimBA [12], Discrete Cosine Transform-based SimBA (SimBA-DCT) [12], and fast surrogate-free black-box attack (Surfree) [27]. Our comparison focuses on the capability of attacks and their robustness, the experimental results are detailed in Table 1.

The experimental data reveal that white-box adversarial examples are robust across different models, suggesting that these examples maintain misdirection effects even on models with significantly different architectures. This observation hints at universally vulnerable decision boundaries. Traditional white-box attacks, however, often overfit a single model, compromising robustness when the target model's structure or data distribution varies from others. To address this, DP-RAE adopts a multi-model strategy to enhance adversarial example transferability and minimize model-specific overfitting, thereby demonstrating improved migration performance across various models compared to conventional white-box attacks.

In black-box attack scenarios, attackers are constrained by their lack of access to the target model's internal mechanisms. While query-based black-box attacks primarily exploit a model's superficial characteristics, they fall short of uncovering deeper vulnerabilities. Notably, black-box attacks generally show limited robustness. Adversarial examples generated by SimBA, SimBA-DCT and Surfree have only achieved single-digit success rates on other models. Our DP-RAE addresses these limitations by initially identifying common vulnerability features across various models through GQBE. Subsequently, it leverages TISA for precise attacks on black-box models, integrating the strengths of both methodologies. DP-RAE has an ASR of over 50% on multiple models, and it achieves a remarkable ASR of up to 94.3% in black-box models under query constraints.

## 4.3 Ablation study

This section presents the ablation studies conducted on DP-RAE to evaluate the impact of different parameters and strategies on its performance. We first focus on the impact of super-pixel size

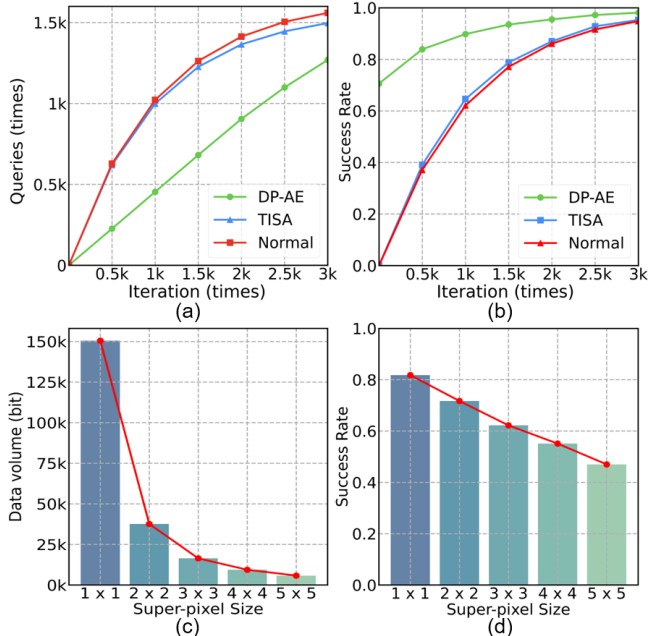

Figure 3: (a) and (b) represent the queries demand and success rate of attacking RN-50, respectively, the normal method represents TISA without historical enhancement. (c) and (d) represent the amount of storage and ASR for different super-pixel sizes, respectively.

on the ASR and the volume of data. To this end, we evaluated the efficacy of adversarial examples crafted with varying super-pixel sizes across different target models. Table 2 shows that adversarial examples created with smaller super-pixel sizes can obtain higher transferability across different models. This is because the effectiveness of adversarial examples depends on the precision and spatial arrangement of the perturbations. As DNNs are highly sensitive to the feature representation of input data, using excessively large super-pixel blocks will oversimplify detailed perturbations. This will reduce their impact on the models' decision boundaries and weaken the attack ability. However, a small size will geometrically increase the number of super-pixels, as shown in Figure 3, the small super-pixel size leads to a dramatic increase in the amount of stored information. Based on the experimental result, we suggest that selecting a super-pixel size of 4 achieves an optimal balance between a high ASR and minimal data storage requirements.

To evaluate the efficacy of GQBE when integrated with various gradient-based adversarial attack methodologies, we employed it with FGSM, BIM, and MI-FGSM, respectively. This experimental setup enabled a comprehensive analysis of how different gradient computation techniques affect the performance of GQBE. According to the data presented in Table 2, iterative methods BIM and MI-FGSM, which calculate perturbations across multiple iterations, outperform FGSM in terms of effectiveness. The iterative refinement of super-pixel perturbations allows for a more precise alignment of adversarial examples with the model's decision boundaries. Notably, MI-FGSM exhibits a marginally superior efficacy to BIM, attributed to the incorporation of momentum, enhancing the

**Table 1: The performance comparison of our method and state-of-the-art adversarial attack on robustness and attack ability. DP-RAE was preprocessed by GQBE through ensemble models: RN-152, Inc-v3, VGG-19.**

| Attack | | ASR (%) on test models | | | | | | | | | |
|---|---|---|---|---|---|---|---|---|---|---|---|
| Method | Target model | RN-34 | RN-50 | DN-121 | Mob-v2 | Mob-v3 | VGG-16 | AlexNet | RN-152 | Inc-v3 | VGG-19 |
| FGSM | VGG-19 | 32.7 | 28.3 | 30.5 | 42.9 | 26.5 | 73.0 | **44.1** | 17.5 | 22.2 | 91.8 |
| FGSM | Inc-v3 | 23.9 | 23.7 | 22.9 | 35.9 | 24.9 | 34.3 | 42.5 | 17.0 | 74.1 | 32.5 |
| FGSM | RN-152 | 35.7 | 37.8 | 32.9 | 38.2 | 23.2 | 39.5 | 43.6 | 76.1 | 25.6 | 38.0 |
| BIM | VGG-19 | 26.0 | 22.8 | 20.6 | 40.3 | 15.9 | **93.7** | 32.1 | 11.8 | 14.6 | **99.7** |
| BIM | Inc-v3 | 22.7 | 19.0 | 21.2 | 28.5 | 18.1 | 27.2 | 30.2 | 11.7 | 97.7 | 27.7 |
| BIM | RN-152 | 48.3 | 61.5 | 43.4 | 41.4 | 17.8 | 40.5 | 34.2 | **99.9** | 25.6 | 41.7 |
| PGD | VGG-19 | 23.6 | 21.3 | 21.3 | 35.0 | 13.7 | 88.7 | 31.1 | 9.9 | 14.0 | 99.7 |
| PGD | Inc-v3 | 21.4 | 18.6 | 19.3 | 29.6 | 17.9 | 27.6 | 31.6 | 11.8 | 95.9 | 26.8 |
| PGD | RN-152 | 44.5 | 58.6 | 41.2 | 36.9 | 18.2 | 40.7 | 31.5 | 99.7 | 25.2 | 38.9 |
| SimBA | RN-50 | 0.5 | 93.8 | 0.5 | 0.7 | 0.2 | 1.6 | 1.3 | 0.3 | 1.0 | 1.6 |
| SimBA-DCT | RN-50 | 0.9 | 86.7 | 0.5 | 1.6 | 0.5 | 1.0 | 1.9 | 0.5 | 1.4 | 0.5 |
| Surfree | RN-50 | 3.3 | 82.0 | 2.8 | 10.5 | 7.4 | 12.7 | 24.5 | 2.4 | 3.5 | 11.4 |
| DP-AE | RN-50 | 56.4 | **95.5** | 56.8 | 52.7 | **38.5** | 66.4 | 42.2 | 94.2 | **98.6** | 96.4 |
| DP-RAE | RN-50 | **56.7** | 94.3 | 56.0 | **53.3** | 38.4 | 66.4 | 41.9 | 94.1 | **98.6** | 96.4 |

**Table 2: ASR (%) of DP-RAE with different settings. The pixel size section demonstrates the ASR in multi-sizes, the embedding attacks section indicates different attack benchmarks applied to GQBE, and the strategy section shows the advantages of dual phase attack.**

| Situation | Pixel Size | | | | | Embedding attacks | | | | Strategy | | |
|---|---|---|---|---|---|---|---|---|---|---|---|---|
| Model | $1 \times 1$ | $2 \times 2$ | $3 \times 3$ | $4 \times 4$ | $5 \times 5$ | FGSM (GQBE) | BIM (GQBE) | MI-FGSM (Smooth) | MI-FGSM (GQBE) | GQBE | TISA | DP-AE |
| RN-34 | **77.9** | 71.4 | 63.3 | 55.8 | 45.2 | 24.0 | 50.3 | 53.3 | **55.8** | 55.8 | 9.0 | **56.4** |
| RN-50 | **81.8** | 71.7 | 62.2 | 55.1 | 47.0 | 19.8 | 49.9 | 52.3 | **55.1** | 55.6 | 91.2 | **95.5** |
| DN-121 | **79.8** | 71.3 | 63.0 | 55.6 | 47.9 | 21.5 | 52.7 | 53.4 | **55.6** | 56.3 | 11.1 | **56.8** |
| Mob-v2 | **79.2** | 69.6 | 62.1 | 52.9 | 47.5 | 27.7 | 46.0 | 50.9 | **52.9** | **54.0** | 10.0 | 52.7 |
| Mob-v3 | 41.9 | 41.2 | **45.6** | 38.9 | 37.0 | 26.7 | 32.1 | 36.7 | **38.9** | 40.5 | 5.4 | 38.5 |
| VGG16 | **94.4** | 84.7 | 79.1 | 67.4 | 53.6 | 28.5 | 64.0 | 64.1 | **67.4** | 68.1 | 8.4 | 66.4 |
| AlexNet | 50.2 | **52.7** | 47.7 | 43.8 | 39.5 | 35.2 | 37.2 | 42.7 | **43.8** | 44.6 | 7.9 | 42.2 |
| RN-152 | **100** | 99.8 | 99.3 | 97.5 | 89.9 | 23.1 | 95.9 | 96.3 | **97.5** | 97.6 | 6.7 | 94.2 |
| Inc-v3 | **99.9** | **99.9** | 99.6 | 98.8 | 98.4 | 53.7 | 98.7 | **99.0** | 98.8 | 98.8 | 7.2 | 98.6 |
| VGG-19 | **100** | 99.9 | 98.7 | 97.4 | 92.3 | 42.8 | 97.1 | 97.1 | **97.4** | 97.5 | 6.8 | 96.4 |

speed in the perturbation calculation process. Therefore, combining GQBE and MI-FGSM for creating adversarial examples greatly improves the attack capability. Additionally, compared to the original MI-FGSM (only smooth), the adversarial examples combined with GQBE demonstrate stronger aggressiveness and robustness, resulting in an improved ASR when attacking different models to varying degrees.

To systematically evaluate the effects of iteration count and the $\mathcal{PCT}$ on the performance of GQBE, a series of experiments were conducted, iterating over a range of $\mathcal{PCT}$ values. The empirical findings, as summarized in Table 3, demonstrate a less number of iterations impedes the perturbation's convergence, leading to diminished attack performance. Consequently, an iteration count of 10 is the optimal setting to ensure adequate convergence while

maintaining computational efficiency. Further examination of the impact of $\mathcal{PCT}$ indicates a performance peak when the parameter is set beyond 50%. This suggests a higher proportion of reinforced pixels contributes to a more effective perturbation strategy, likely due to the enhanced potential for inducing misclassifications within the target model. Therefore, we advocate for a $\mathcal{PCT}$ setting of 50%, representing the threshold at which additional increments cease to yield proportional gains in attack performance.

To evaluate the hypothesis that a dual-phase strategy—integrating both black-box and white-box attacks—can leverage the strengths of each approach, we designed ablation studies focusing on three distinct strategies: solely employing GQBE, solely employing TISA, and a combined DP-AE targeting RN-50. For TISA, we fixed the

**Table 3: The ASR (%) of GQBE with different iteration count and PCT settings.**

| Iteration : 5 steps | Test model | | | | | | | Attack model | | |
|---|---|---|---|---|---|---|---|---|---|---|
| Attack | RN-34 | RN-50 | DN-121 | Mob-v2 | Mob-v3 | VGG-16 | AlexNet | RN-152 | Inc-v3 | VGG-19 |
| GQBE ($\mathcal{PCT} = 0$) | 46.7 | 46.3 | 48.5 | 44.8 | 36.4 | 57.0 | 41.6 | 86.4 | 97.2 | 93.7 |
| GQBE ($\mathcal{PCT} = 0.1$) | 48.5 | 49.0 | **51.5** | 48.6 | 36.4 | 60.8 | 43.0 | 88.9 | 97.5 | 94.2 |
| GQBE ($\mathcal{PCT} = 0.3$) | **50.4** | 49.1 | 50.1 | 49.6 | 38.0 | 61.7 | 43.0 | **89.9** | 97.9 | 94.8 |
| GQBE ($\mathcal{PCT} = 0.5$) | 50.0 | 49.6 | **51.5** | **50.6** | 38.8 | 62.9 | **44.0** | 89.6 | **98.0** | **95.0** |
| GQBE ($\mathcal{PCT} = 0.7$) | **50.4** | **50.1** | 51.3 | 49.5 | **39.2** | **63.1** | 43.6 | **89.9** | 97.8 | 94.4 |
| **Iteration : 10 steps** | **Test model** | | | | | | | **Attack model** | | |
| Attack | RN-34 | RN-50 | DN-121 | Mob-v2 | Mob-v3 | VGG-16 | AlexNet | RN-152 | Inc-v3 | VGG-19 |
| GQBE ($\mathcal{PCT} = 0$) | 53.1 | 52.3 | 53.3 | 51.0 | 36.9 | 64.0 | 42.7 | 96.3 | **99.0** | 97.1 |
| GQBE ($\mathcal{PCT} = 0.1$) | 54.8 | 54.8 | 53.8 | 53.4 | 38.1 | 66.4 | 44.2 | 96.6 | **99.0** | 97.5 |
| GQBE ($\mathcal{PCT} = 0.3$) | 55.7 | 55.5 | 56.0 | 52.6 | 39.0 | 67.0 | 44.1 | 97.3 | 98.8 | **97.6** |
| GQBE ($\mathcal{PCT} = 0.5$) | **55.8** | 55.6 | 56.3 | 54.0 | **40.5** | **68.1** | 44.6 | **97.6** | 98.8 | 97.5 |
| GQBE ($\mathcal{PCT} = 0.7$) | 55.3 | **57.2** | **56.8** | **54.9** | 40.0 | 68.0 | **45.6** | 97.0 | **99.0** | 97.3 |

iterations at 3000, with a history recorder size set to 10. The empirical results, depicted in Figure 3, indicate that DP-AE not only achieves a superior ASR but also necessitates fewer queries than the standalone strategies. This improvement can be attributed to the preprocessing phase of the white-box attack, leveraging its inherent robustness. White-box attacks can effectively reduce the confidence of certain examples in unknown models and even misclassify some examples before the commencement of black-box attacks, thereby significantly reducing the need for extensive queries in the subsequent black-box attack phase. Strategically incorporating historical data further reduces query count while bolstering overall attack efficacy.

Subsequent robustness tests across the three strategies reveal distinct performance profiles in adversarial example generation. As Table 2 shows, GQBE produced adversarial examples exhibit commendable generalization across diverse models yet fall short of achieving an optimal ASR against the targeted RN-50 model. Conversely, TISA generated samples demonstrate a high degree of specificity in compromising unknown models but suffer from limited transferability, undermining their robustness. Notably, adversarial examples generated via the DP-AE strategy not only excel in model-specific attacks but also benefit from enhanced transferability, showcasing an advantageous blend of robustness and specificity. Our findings thus advocate for the DP-AE approach, which embodies a harmonious balance between achieving targeted model vulnerability and ensuring broad applicability across various model architectures.

## 4.4 Robustness evaluation

In real-world scenarios, images often undergo preprocessing via defensive mechanisms before being fed into neural networks, aiming to retain critical content while reducing the impact of perturbations. Hence, the necessity for adversarial examples to exhibit considerable robustness becomes paramount. This section explores the effects of various defense strategies on adversarial examples, including Spatial Squeezing (Spatial) [5], Random Resizing and Padding (Random) [42], Gaussian Blurring (Gaussian) [1], JPEG

Compression (JPEG) [5], and Super-resolution (Super) [29]. As depicted in Table 4, perturbations from black-box attacks tend to lose their potency under defensive measures, likely due to preprocessing adding uncertainty to the model and thus diluting the effectiveness of query-based attacks. Conversely, DP-RAE employs the white-box preprocessing technique, GQBE, which identifies and exploits common vulnerabilities across models, diminishing the impact of defensive methods. By integrating multiple models, DP-RAE addresses the overfitting issue and enhances resilience against defensive tactics. As a result, DP-RAE showcases exceptional adaptability and robustness, maintaining high attack success rates against diverse defense strategies.

**Table 4: The ASR (%) of adversarial attacks when againsting different defence methods.**

| Attack method | Defense method | | | | |
|---|---|---|---|---|---|
| | Spatial | Random | Gaussian | JPEG | Super |
| FGSM [11] | 55.9 | 53.4 | 48.2 | 62.6 | 75.1 |
| BIM [20] | 51.5 | 62.3 | 38.6 | 60.2 | **90.7** |
| PGD [26] | 49.4 | 57.1 | 37.7 | 56.3 | 87.2 |
| SimBA [12] | 18.7 | 3.8 | 16.7 | 4.3 | 2.0 |
| SimBA-DCT [12] | 19.5 | 3.9 | 15.9 | 3.5 | 1.7 |
| Surfree [27] | 19.6 | 6.7 | 17.7 | 5.2 | 3.5 |
| DP-AE (Ours) | 63.1 | **62.9** | **59.4** | **69.8** | 83.6 |
| DP-RAE (Ours) | **63.2** | 62.6 | 59.2 | 69.4 | 82.3 |

## 4.5 Reversibility of DP-RAE

The restoration effect of adversarial images is pivotal in underscoring DP-RAE's efficacy. Notably, prior works have not extensively explored the recovery performance of RAEs. To validate the restoration capabilities of RAEs, we compared images restored by DP-RAE with their original counterparts. As depicted in Figure 4, while adversarial manipulations may marginally impact visual quality, they do not hinder the accurate recognition of the content by human observers. The restoration process effectively neutralizes DP-RAE's

 

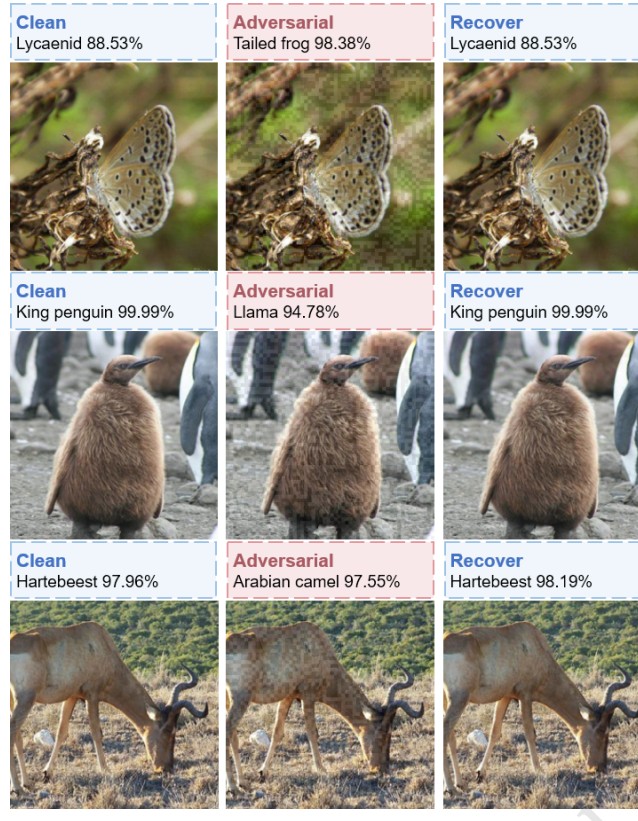

**Figure 4: Visual effects before and after DP-RAE recovery and the ability of DP-RAE to mislead the model.**

perturbations, thereby not only restoring visual fidelity but also recovering the neural network's initial classification accuracy for these examples. Table 5 evaluates the image quality post-DP-RAE restoration under three distinct perturbation intensities, revealing that the Post-Recovery PSNR for all perturbation levels exceeds 40 dB and the SSIM approaches 1. This indicates exceptional restored image quality. Moreover, the misclassification rate for images post-recovery through DP-RAE significantly drops from 94.3% to 0%, affirming DP-RAE's capacity to effectively counteract perturbations that compromise the model's classification accuracy, thus facilitating self-recovery.

**Table 5: The recoverability of our attack method varies with different levels of unit noise. "Adversarial" and "Recover" respectively indicate whether the detected image is an adversarial example or a recovered image, "↑" means the bigger the better.**

| Unit noise | Adversarial | | | Recover | | |
|---|---|---|---|---|---|---|
| | PSNR ↑ | SSIM ↑ | ASR ↑ | PSNR ↑ | SSIM ↑ | ASR ↓ |
| 3/255 | **30.42** | **0.8415** | 89.9 | **44.40** | **0.9877** | **0** |
| 4/255 | 27.94 | 0.7667 | 94.3 | 43.61 | 0.9841 | **0** |
| 5/255 | 25.97 | 0.6960 | **96.9** | 42.76 | 0.9801 | **0** |

**Figure 5: In the commercial model, clean image identified as a "Rocking chair", DP-RAE misclassified as "Historic sites".**

## 4.6 Commercial model attack

To validate our RAE's effectiveness on real-world systems, we targeted Baidu's cloud vision API[1], a service for object recognition. We aimed to eliminate the top label from the API's top-3 returned labels, considering the constraints on perturbations and queries. We prepared 50 images, correctly identified by the API, for enhanced robustness through white-box attacks. These attacks successfully misled 70% of the images, thanks to significant perturbations, which are reversible in RAEs, minimizing the impact on legitimate users. Subsequent black-box attacks on the rest, by probing the API for decision boundaries, achieved a 90% success rate.

As shown in Figure 5, we achieve the model misclassify from the original label "rocking chair" to the wrong label "historic sites", underscoring our method's threat to commercial black-box models and its role in protecting user privacy. Given the limitation on the number of queries allowed for commercial models, we believe that increasing the number of queries can effectively enhance the success rate of the attack. **All relevant details and the corresponding analyses of the evaluation results are included in our provided supplemental material.**

## 5 CONCLUSION

Based on the harm posed by adversarial examples to deep neural networks, we leverage this characteristic as a new mechanism for social privacy protection. This work introduces the DP-RAE framework for creating robust adversarial examples aimed at specific black-box models. Through initial white-box attack preprocessing, these examples become more robust and simplify subsequent black-box attacks. Utilizing historical query data, our heuristic black-box attacks improve efficiency. Our experiments demonstrate that DP-RAE surpasses conventional ones in effectiveness. Moreover, DP-RAE marks the inception of reversible adversarial examples applicable to commercial black-box models, combining recoverability and robustness to offer a novel privacy protection solution.

[1]https://ai.baidu.com/tech/imagerecognition/general

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
