# OpenReview forum: "DP-RAE: A Dual-Phase Merging Reversible Adversarial Example for Image Privacy Protection"
_acmmm.org/ACMMM/2024/Conference — MM2024 Poster_

### Official Review · Reviewer_TSRB · 2024-05-15

**Rating:** 3
**Confidence:** 4

**Summary:**

This paper proposes a dual-phase merging reversible adversarial example method for image privacy protection by combining a heuristic black-box attack and reversible data hiding with grayscale invariance. The author conducts extensive experiments, including the effectiveness of adversarial attacks and data hiding, to demonstrate the effectiveness of their method.

**Strengths:**

1. The author uses adversarial examples to perform privacy protection, which enriches the practical use of adversarial examples. However, a more realistic scenario should be considered to verify its useless, for example, a facial recognition system.
2. The author conducts extensive experiments to evaluate the effectiveness of their method.

**Limitations:**

1. The writing and logic of the paper needed improvement. In the abstract, lack of a clear introduction to their proposed method. Moreover, the author spends a lot of text introducing their attack method while a little introduction to the data embedding and recovery.
2. I wonder to know, how to calculate the gradient with respect to super-pixels, as the dimension expansion and padding are not differentiable.
3. Lack of some necessary ablation study, for example, 1) How to determine the percentage of PCT, and how does it influence the attack performance? 2) How to determine the weight of wk for each ensemble model?
4. The discussion about the embed and recovery is limited. Does the embedded data degrade the attack performance? And how to ensure the recover accuracy when the adversarial example with secret data is distorted.
5. There are some typo errors, for example, the notion of y in line 274.
6. As reported in Table 1, the comparison of adversarial attacks is unfair; the author adopts four models to perform an ensemble attack, while the comparison method is to perform a single model attack.
Moreover, as the baseline of adversarial attack, the comparison method is to old, latest baseline should be considered, for example, admix[1], ssifgsm[2], and cwa [3].
[1] Xiaosen Wang, Xuanran He, Jingdong Wang, and Kun He. Admix: Enhancing the transferability of adversarial attacks. InProceedings of the IEEE/CVF International Conference on Computer Vision, pages 16158–16167, 2021.
[2] Long, Yuyang, et al. "Frequency domain model augmentation for adversarial attack." European conference on computer vision.Cham: Springer Nature Switzerland, 2022.
[3] Chen, Huanran, et al. "Rethinking Model Ensemble in Transfer-based Adversarial Attacks." The Twelfth International Conference onLearning Representations. 2024.

**Suitability:**

3

---

### Official Review · Reviewer_2B3r · 2024-05-19

**Rating:** 4
**Confidence:** 2

**Summary:**

The paper introduces a novel framework called Dual-Phase Merging Reversible Adversarial Example (DP-RAE) which integrates adversarial attacks and Reversible Data Hiding (RDH). This is aimed at thwarting unauthorized access while maintaining image quality.

**Strengths:**

* The paper presents a unique integration of adversarial attacks and reversible data hiding, addressing the dual needs of effective adversarial manipulation and complete image recovery.

* Experimental results are impressive, with a high attack success rate and high quailty of the recovery, substantiating the method’s effectiveness.

**Limitations:**

1.The method proposed in this article should be compared with other RAE methods since this article focuses on Reversible Adversarial Examples (RAE), instead of only considering traditional adversarial attacks as the baseline.

2. According to the visual results presented in Figure 4, it appears that the adversarial examples (DP-RAE) introduce perceptible perturbations that could be easily detected by humans. Furthermore, the quantitative image quality results presented in Table 5 are difficult to assess due to a lack of comparisons with other methods.

3. Questions regarding the technical details: What is the difference between \eta and \xi in the GQBE algorithm? Could you provide more insight or experimental justification for the particular selection of the three points that exhibit the most substantial decrease (Sec 3.3)?

**Suitability:**

2

---

### Official Review · Reviewer_EbyV · 2024-05-25

**Rating:** 5
**Confidence:** 3

**Summary:**

The paper presents a novel dual-phase merging framework called Dual-Phase Merging Reversible Adversarial Example (DP-RAE) for RAE generation, significantly enhancing the transferability of adversarial examples. The framework aims to protect image privacy on social networks by preventing unauthorized access and analysis by DNNs, while maintaining image quality for authorized users.

**Strengths:**

1.	The paper introduces a dual-phase strategy that merges heuristic black-box attacks with RDH technology, ensuring both effective adversarial attacks and the complete recovery of the original image.
2.	The paper introduces innovative perturbation optimization techniques like Gradient Quantized Binary Encoding (GQBE) for white-box scenarios and Threshold-Informed Superpixel Attack (TISA) for black-box contexts, which maintain the integrity of adversarial examples while ensuring their reversibility.
3.	The evaluation of the paper covers multiple aspects of the framework’s performance. DP-RAE shows superior performance compared to other methods, with high ASR across various models, especially against black-box models under query constraints.
4.	The proposed framework achieves a 100% restoration rate for perturbed images, indicating that the original image can be perfectly restored after the attack.
5.	The paper is clearly structured and easy to read.

**Limitations:**

1.	Although the paper claims a high success rate and recovery rate, it may benefit from a more comprehensive evaluation, including a wider range of datasets and an analysis of potential failure cases. The authors can also try more combinations of different target models and ensemble models.
2.	The dual-phase approach and the integration of different attacks and RDH module may result in increased computational complexity, potentially affecting the efficiency of the method.
3.	The attack described in the paper interferes with all pixels of the image, which may reduce image quality and make it easily detectable.
4.	The paper indicates that smaller super-pixel sizes achieve higher ASR, but they also lead to a significant increase in the amount of stored information. The authors could attempt to balance the ASR and data storage by limiting the size of the perturbation-covered image area.

**Suitability:**

3

---

### Official Review · Reviewer_Sg93 · 2024-05-25

**Rating:** 5
**Confidence:** 3

**Summary:**

This research proposes a two-stage merging framework for generating RAEs (reversible adversarial samples) and introduces two novel perturbation optimization techniques: gradient-quantized binary encoding (GQBE) for white-box scenarios and threshold-aware superpixel attack (TISA) for black-box environments, thus solving the problem of perturbation storage posed by RDH (reversible data hiding).The DP-RAE framework aims to Create robust adversarial samples against specific black-box models, which are made more robust by initial white-box attack preprocessing and simplify subsequent black-box attacks. Combining heuristic black-box attacks with RDH-GI (Reversible Data Hiding with Gray Scale Invariance) techniques not only maintains the effectiveness of the adversarial attacks, but also ensures the reversibility of the process, which allows for the seamless recovery of the unmodified original image. In addition, this research validates a realistic system for the first time.

**Strengths:**

The DP-RAE framework introduces a biphasic approach combining white-box and black-box attack techniques, enhancing the portability and robustness of the adversarial samples. A solid theoretical foundation is demonstrated by optimizing perturbations using GQBE and TISA. Detailed algorithmic descriptions and mathematical formulations validate the correctness of the techniques. The effectiveness of the framework is demonstrated in extensive experiments on a variety of models, achieving a 96.9% attack success rate and 100% recovery rate. Ablation studies and comparisons with state-of-the-art methods comprehensively demonstrate the superiority of the framework. Furthermore, DP-RAE marks the first application of reversible adversarial samples in a commercial black-box model, combining recoverability and robustness to provide a novel privacy-preserving solution.

**Limitations:**

The paper does not discuss in depth the limitations and potential drawbacks of the proposed method, such as computational complexity or impact on image quality in real-time applications.
While the paper compares DP-RAE with several counter-attack methods, a more in-depth comparison with more recent state-of-the-art techniques could have enhanced the evaluation.
The sensitivity of the framework to various parameter settings could be explored in more detail to better understand its robustness and adaptability.

**Suitability:**

3

---

### Meta-Review · Area_Chair_MUS9 · 2024-07-04

**Recommendation:** Accept (Poster)
**Confidence:** 5

**Metareview:**

This paper presents a new dual-phase merging framework called Dual-Phase Merging Reversible Adversarial Example (DP-RAE) for RAE generation. The reviewers unanimously recommend accepting the paper. However, the authors are urged to address Reviewer EbyV's concerns in their final version.

---

### Meta-Review · Senior_Area_Chairs · 2024-07-10

**Recommendation:** Accept (Poster)
**Confidence:** 4

**Metareview:**

All the reviewers gave positive ratings and tend to accept the paper. SAC and AC agree with reviewers and recommend acceptance of the paper.